# Brassinosteroid Promotes Grape Berry Quality-Focus on Physicochemical Qualities and Their Coordination with Enzymatic and Molecular Processes: A Review

**DOI:** 10.3390/ijms24010445

**Published:** 2022-12-27

**Authors:** Jiajia Li, Yi Quan, Lei Wang, Shiping Wang

**Affiliations:** Department of Plant Science, School of Agriculture and Biology, Shanghai Jiao Tong University, Shanghai 200240, China

**Keywords:** brassinosteroid, fruit quality, grapevine, phytohormones interaction

## Abstract

Brassinosteroid (BR) is an important endogenous phytohormone that plays a significant role in fruit quality regulation. The regulation of BR biosynthesis and its physiological effects have been well-studied in various fruits. External quality (fruit longitudinal and transverse diameters, firmness, single berry weight, color) and internal quality (sugars, aroma, anthocyanin, stress-related metabolites) are important parameters that are modified during grape berry development and ripening. Grapevines are grown all over the world as a cash crop and utilized for fresh consumption, wine manufacture, and raisin production. In this paper, the biosynthesis and signaling transduction of BR in grapevine were summarized, as well as the recent developments in understanding the role of BR in regulating the external quality (fruit longitudinal and transverse diameters, firmness, single berry weight, and color) and internal quality (sugars, organic acids, aroma substances, anthocyanins, antioxidants) of grapes. Additionally, current advancements in exogenous BR strategies for improving grape berries quality were examined from the perspectives of enzymatic activity and transcriptional regulation. Furthermore, the interaction between BR and other phytohormones regulating the grape berry quality was also discussed, aiming to provide a reliable reference for better understanding the potential value of BR in the grape/wine industry.

## 1. Introduction

As a new plant endogenous hormone, BL, also acknowledged to be a kind of BRs, has the potential to significantly increase plant growth and promote fertilization at low concentrations [1], as well as play a role in increasing chlorophyll content, improving photosynthesis efficiency, promoting root development and seedling, blooming, and maturation [2,3,4]. Meanwhile, it could be responsible for improving stress tolerance in plants, such as cold, drought, salt, and acid resistance [5]. At present, BL is acknowledged as the sixth largest phytohormone, even though it presents in small amounts as compared to other five phytohormones (auxin, abscisic acid (ABA), cytokinin, gibberellin, and salicylic acid (SA)), but it is more effective in term of growth and development.

Fruit quality, as an important parameter to reflect the growth and development of fruits, could be divided into two categories: external and internal quality. Fruit external quality, including fruit longitudinal and transverse diameters, firmness, and single berry weight, has been demonstrated to be influenced by exogenous BL in both climacteric fruit and non-climacteric fruit [6]. Fruit internal quality is determined by biochemical characteristics, which are important indicators of fruit flavor superiority or inferiority. Flavor is also an essential indicator for determining the condition of fruit development and ripening, and it has been proven that BR is strongly linked to the formation of fruit flavor [7]. Fruit flavor is produced by metabolic changes in sugar, acid, amino acids, etc. Sugars have an important role in the production of fruit flavor, as evidenced by the fact that endogenous BL levels are favorably associated with sugar levels in rice [8], as well as immensely promoting the accumulation of sugars in citrus [9]. Acidity is also an important determinant of fruit flavor, and it has been proven that exogenous application of 24-epibrassinolide (EBL, brassinolide-like plant growth regulator) greatly reduces acidity under conventional cultivation mode, thus maintaining fruit quality [10]. In the late stages of fruit development, amino acids are abundant, and they play an important role in the production of fruit flavor. Amino acid anabolism and BR signal transduction were found to have crosstalk, and this interaction lead to a considerable improvement in fruit quality [11]. Aroma is another important indicator of fruit ripeness and edibility. According to the previous study [12], BR spraying on grapevines greatly increases the biosynthesis of terpene compounds in grape berries at the maturity stage, and speeds up the enrichment of aroma components. Moreover, after the application of BL to “Shine Muscat” grape berries, not only was the content of linalool considerably increased, but the proportion of methyl glyoxal, pentanal, and hexanal were developed significantly [13]. However, fatty acids are the most important indicator of fruit nutritional content, and their synthesis interacts with BR signaling transduction in a significant way [14]. Exogenous BL was found to increase the concentration of fatty acids in tomato fruits, leading to an improved taste and nutritional value [15].

BR interaction with other phytohormones has been studied well in recent years. The interaction of BR-indole-3-acetic acid (IAA) regulated the maintenance of root apical meristem in the model plant [16], and the spraying of IAA and BR significantly improved the antioxidant content and fruit quality of apple fruits [17]. There was also an interaction between ABA and BL, indicating that the combined effect of ABA and BL application improved drought tolerance and photosynthesis in tall fescue [18] and blueberry quality [19]. Moreover, gibberellins, BL, and ethylene (ETH) signaling were involved in flower differentiation and development [20]. The interaction among BR, IAA, and gibberellins triggers the growth of apple trees [21]. Despite this, the exogenous application of SA and EBL improves antioxidant capacity and secondary metabolites in Brassica [22] and played a significant role in the plant defense system improvement in Arabidopsis as well [23]. Cytokinins are responsible for regulating plant growth and development, and their contents were also dramatically increased under exogenous BL treatment [24]. The findings above clearly supported the notion that there was an interaction between many phytohormones and BR signaling in plants, whereas the relationship in grapes has to be investigated further.

Grapevine has become one of the world’s most important horticulture crops, with a planted area of more than 7.8 million hectares, and is utilized for a wide range of products, including fresh table grapes, juice, preserves, raisins, and wine [25]. It has been acknowledged that phytohormones such as ABA, IAA, and cytokinins played an important role in grapevine development and ripening, and corporately contribute to regulating the improvement of fruit quality. The sixth plant hormone “BR” has also made some progress in terms of understanding its role in grape berry quality regulation. Therefore, the objective of this review was to describe the BR biosynthesis and signal transduction systematically, as well as to characterize the influence of BR on both the external and internal quality of grape berries and examine the interaction at the molecular level. Moreover, the progress made in understanding the interaction between BR and other phytohormones had also been reviewed, hoping to provide references for further understanding of the physiological effects of BR and its effect on fruit quality regulation.

## 2. Regulation of BR Biosynthesis and Signal Transduction

It has been well-elucidated that BR biosynthesis was regulated by a series of structural genes and transcription factors. The conversion of cycloartenol to campesterol sterol, followed by the conversion of campesterol sterol to BR, is the first step in the biosynthesis of BR. The C-6 early and C-6 late are the two primary pathways for BR biosynthesis, and they both require a set of regulatory genes and catalytic enzymes. Several upstream genes *dwarf4* (*DWF4), deetiolated2 (DET2), constitutive photomorphogenesis and dwarfism (CPD),* and *3-epi-6-deoxocathasterone 23-monooxygenase* (*ROT3),* and downstream genes *brassinosteroid-6-oxidase1,2* (*BR6OX1, BR60X2*) were responsible for BR-biosynthesis. Furthermore, PhyB activation-tagged suppressor1 (BAS1) is responsible for BR metabolites and plays an essential role in the stability of BR content in plants [26]. The specific BR biosynthesis and the metabolic pathway are present in Figure 1. On the cell surface of *Arabidopsis thaliana*, researchers discovered three BR receptors: BRASSINOSTEROID-IN-SENSITIVE 1 (BRI1), Bri1-like 1 (BRL1), and Bri1-like 3 (BRL3) [27]. The activities of both proteins (BRI1 heterodimer and trans-phosphorylation of BRI1-associated kinase 1) were enhanced by BR signaling [28]. When BR levels are high, a cascade of phosphorylation in the BRI1 and BAK1 (Brassinosteroid Insensitive 1 associated kinase receptor 1) proteins was initiated. Subsequently, BSU1/BSLs (BRI1SUPPRESSOR 1) were dephosphorylated and generated BRASSINOSTEROID-INSENSITIVE 2 (BIN2), which was then further inactivated [29]. Furthermore, BES1/BZR1 was dephosphorylated and activated by protein phosphatase 2A (PP2A), which is recognized as a BR signaling positive regulator [30]. In case of a low level of BR in plants, the BR BRI1KINASE INHIBITOR 1 (BKI1) occupies the BRI1 site, and inhibits the interaction of bri1-BAKI1 [31]. Following that, subsequent phosphorylation cascade was repressed, BSU1/BSLs (BR signal receptor protein) were deactivated by dephosphorylation, leading to the activation of BIN2 (Bridging integrator 2) and further transmission of BR signal. In addition, phosphorylated BES1/BZR1 (BRI1 EMS SUPPRESSOR 1/BRASSINAZOLERESISTANT 1) in the cytoplasm was degraded by Ubiquitin Proteasome System (UPS), thus terminating BR signaling [32]. In conclusion, the primary regulatory mechanisms of BR signal transduction were phosphorylation/dephosphorylation and ubiquitination, which are essential components in the BR signaling pathway and also regulated their activity and stability. Overall, the BR signaling pathway has been thoroughly investigated; the unique BR signaling pathway is depicted in Figure 2.

## 3. Effect of BR on Grape Berry Development and Quality Improvement

BR has a considerable impact on the quality of grape berries. The influence of BR on external fruit quality parameters, biochemical characteristics, and the involvement of enzymes and genes to increase grape aroma, antioxidant activity, color, and other biochemical aspects were summarized in this review. On the basis of earlier research, we created a model diagram of exogenous BR regulating grapefruit quality enhancement, as shown in Figure 3.

### 3.1. Effect of BR on Table Grape External Quality Development

Grapevine external quality, including berry weight, berry size, fruit firmness, fruit color, and cluster weight, is an important parameter to reflect fruit development and fruit quality. The fruit with a smooth texture and gorgeous color is more attractive to consumers and has a better market development prospect [33]. Generally speaking, the first criterion for consumers is whether the grape berries are fully purplish-red and uniform in size [34].

Numerous reports confirmed that the application of exogenous phytohormones can significantly improve the grapes’ external quality (fruit longitudinal and transverse diameters, firmness, single berry weight and color), in which BR plays an important role in this process [35]. For instance, applied different concentrations of 24-epibrassinosteroid (EBR, brassinolide-like plant growth regulator) (0.2, 0.4, 0.6, and 0.8 mg L^−1^) were as an exogenous application on the nine-year-old grapevine “Alphonse Lavallée” at different berry developmental stages. In both growing years, compared with the control group and other BR concentrations, the application of 0.2 mg L^−1^ of BR to vines three times (7 days after berry set+veraison + 30 days after veraison) provided the maximum berry weight (7.65 g and 7.87 g), cluster weight (374.98 g and 418.75 g), and yield production (26.25 vine kg^−1^ and 30.15 vine kg^−1^) [36]. Studies studied the postharvest effect of BR at the concentrations of 0, 0.75, and 1.5 ppm on one-year-old “Rish Baba” grapevine and identified that 1.5 ppm treatments were effective and responsible for reducing single fruit weight from 29.75% to 29.48% under cold storage for 5 weeks [37]. Furthermore, BR was applied on a twelve-year-old “Flame Seedless” grapevine and it was found the berry weight, berry length, and berry breadth were higher at 0.5 ppm (2.55 g, 1.84 cm, and 1.73 cm, respectively) and 1 ppm (2.57 g, 1.89 cm, and 1.74 cm, respectively) as compared to control (2.38 g, 1.71 cm, and 1.59 cm, respectively) [38]. A previous study used BR and ABA on mature eight-year-old “Shine Muscat”, as well as their combination (BR + ABA), and samples were taken after 12, 24, 36, and 48 h of treatment. Regardless of treatments, the weight of shine Muscat was non-significant at all sampling times. At 12, 24, 36, and 48 h, the BR treatment gained the weight (69.39, 69.18, 68.99, and 68.89, respectively), followed by BR + ABA (65.87, 65.65, 65.46, and 65.39, respectively), control (65.77, 65.55, 65.30, and 65.23, respectively), and ABA (65.74, 65.53, 65.33, and 65.26, respectively) [13]. Exogenous application of 28-homobrassinolide (4 ppm) decreased fruit firmness (6.01 N) in 12-year-old “Sultani” grapes more than the control group (6.19 N) [39]. Researchers applied the different concentrations of EBL (0.1, 0.4, 0.8 mg L^−1^) as an exogenous application on five-year-old “Redglobe” grapes, and the fruit firmness was decreased after 15 days (5.25 N), 30 days (5.15 N), and 60 days (4.98 N) of storage under 0.8 mg L^−1^ EBL treatment [40].

In terms of the effect of BR on the color change of grape berries, exogenous BR (3-hydroxy-20-RB-homo-7-oxa-5-cholestan-6-one) application of 0.4 mg L^−1^ significantly contributed to the CIRG (the color parameter of a red grape variety, as determined by the CIELAB parameters *L** (brightness), H (tone angle), and C (color temperature) (chroma)) of sixteen-year-old self-rooted “Redglobe” grapevine [41]. The application of BRs (0.5 and 1.0 mg L^−1^) on the twelve-year-old “Flame Seedless” grape cluster effectively delayed the deviation rates of *L**, *A**, and *B**, and was responsible for color changing from relatively pure green to yellow and subsequently to a red color when compared to the lowest dose (0.1 mg L^−1^ BR) and the control [38]. The study also found that a 100 μmol L^−1^ BR treatment improved five-year-old “Kyoho” grape pericarp coloring at various phases of fruit development, with the most noticeable effect occurring at the start of veraison [12]. According to the aforementioned results, exogenous BR significantly affected berry color, fruit firmness, fruit expansion, and yield development. As a result, a brief overview of the exogenous BR study on grape external quality was compiled in Table 1. Further research is required to understand the molecular mechanisms by which BR enhances the different physical attributes of grapes.

### 3.2. Effect of BR on Grape Biochemical Properties

The biochemical properties of grape berries could be reflected by aroma, sugar and acid ratio, soluble solid content, nutrient composition, storage, and transportation performance, etc. [48]. In recent years, exogenous phytohormones have been shown to promote biochemical properties in grape berries, in which BR plays a key role in sugar accumulation, organic acid enrichment, and phenolic synthesis [49]. The research progress on the effect of exogenous BR on the biochemical properties formation of grapes was enumerated in Table 2 in order to better sort out the relationship between exogenous BR and biochemical properties.

#### 3.2.1. Role of Biochemical Properties on Fruit Flavor

Biochemical features and fruit quality development are greatly influenced by the accumulation of metabolites connected to flavor synthesis. Additionally, flavor development is dependent on a balanced mix of sweet, sour, and astringent qualities in the fruit [57]. The sugars in fruit are mostly responsible for sweetness. Sugars in grapes fruits accumulate to a very high level as a major metabolite and energy source with fruit ripening, and reach a peak in the late maturity stage, directly reflecting the fruit harvestability and maturity [58].

#### 3.2.2. Effect of BR on Sugars Accumulation in Grape Berries

BR has been involved in the regulation of sugar accumulation in grape berries in a series of studies. Three different concentrations of EBL (0.1 mg L^−1^, 0.4 mg L^−1^, and 0.8 mg L^−1^) were used to treat the “Cabernet Sauvignon” and “Yan 73” berries and the outcome of this study revealed that 0.4 mg L^−1^ significantly enhanced the total soluble solids, reducing the sugar content in Cabernet Sauvignon (19 °Bx and 167 g L^−1^, respectively) and Yan 73 (18 °Bx and 165 g L^−1^, respectively) as compared to other EBL concentrations and control [46]. 0.6 mg L^−1^ EBL treatment significantly enhanced the accumulation of monosaccharides in ten-year-old “Merlot” grape berries, including an increase in glucose and fructose content (16.95% and 39.31%, respectively) at the maturity stage, compared to the control [50]. Another study validated the same experimental results using the twelve-year-old “Thompson Seedless” grape as test material, finding that 3 and 6 mol L^−1^ EBL significantly increased TSS levels (22 °Bx and 22.5 °Bx, respectively) as compared to the control group (18.5 °Bx) [51]. Furthermore, using different concentrations of EBR as foliar sprayed on the eighteen-year-old “Khalili” grape, the researcher verified that 0.4 mg L^−1^ EBR significantly increased the total soluble solid content of (22.26 °Bx), which was higher than that of 0.2 mg L^−1^ EBR treatment (21.33 °Bx) and the control group (18.94 °Bx) [42].

Studies at the enzymatic level revealed that exogenous BR also played an important role in promoting glucose accumulation, including controlled sugar unloading in five-year-old “Cabernet Sauvignon” berries during veraison. Using 0.4 mg L^−1^ EBL treatment in grape pericarp from days after application (DAA) 85 to 100, the glucose and fructose conversion enzymes “acidic invertase (INV) and neutral invertase (NI)” significantly up-regulate their activity, while the 1.31 mg L^−1^ Brz (BR signaling inhibitor) application significantly reduced the acidic invertase (INV) activity and the INV activity at 60 DAA and 66 DAA, respectively [44]. In addition, compared with the control group, using 0.6 mg L^−1^ exogenous EBL two folds enhanced the activity of sucrose phosphate synthase (SPS), significantly up-regulated activities of cell wall acid invertase (VvcwINV), sucrose transporter (VvSUC12), and sucrose synthase (VvSS) in ten-year-old “Merlot” grape from veraison to ripening stage [50].

According to studies, exogenous BR seems to play a key function in increasing glucose accumulation at the molecular level. The overexpression of *VvSK7* (Glycogen synthase kinase 3 family gene) in Arabidopsis plants regulates glucose accumulation in response to BR signaling, and the application of 1.5 μmol L^−1^ of EBR and 1.5 μmol L^−1^ of EBR + 70 μmol L^−1^ of BIKININ (GSK3/Shagy-like kinase inhibitor) on five-year-old “Shine Muscat” significantly promotes soluble solids accumulation by inhibition of glycogen synthase kinase 3/shaggy kinase-like (*VvSKs*) expression [59]. Moreover, 0.4 mg L^−1^ of EBL treatment on the five-year-old “Cabernet Sauvignon” grape increased the transcription levels of hexose transporters, which were sugar transporters (*VvHT3*, *VvHT4*, *VvHT5*, and *VvHT6)* at various stages of berry development, including half veraison stage and maturity stage, but had little effect on hexose transporters (*VvHT1* and *VvHT2*) expressions, whereas Brz (BR signal transduction inhibitors) and Brz + EBL treatment down-regulated hexose transporter1 *(VvHT1*) transcription levels at 66 DAA. The transcription level of VvSUC27 (sugar transporters) was considerably greater in grape berries treated with 0.4 mg L^−1^ EBL and 1.31 mg L^−1^ Brz + 0.4 mg L^−1^ EBL from 60 to 93 DAA than in grape berries treated with Brz [44]. Thus, BR was reported to have a considerable impact on sugar accumulation; however, a more in-depth study is required to properly understand the mechanisms underlying the interaction between the sugar signal and the BR signal.

#### 3.2.3. Effect of BR on Organic Acids Formation in Grape Berries

Acidity is another important biochemical attribute that influences fruit flavor. As fruit ripens, organic acids degrade into other organic matter, and the contents of total acid are usually inversely proportional to total soluble solids during fruit maturation [60]. The most prevalent organic acids in grape grapes were malic acid and tartaric acid, and their synthesis was regulated by exogenous BR over many years, In the first year, the highest TA was obtained through the use of 0.2 mg L^−1^ of BR at 7 days after fruit set  +  veraison (6th application) (*p*  <  0.0001), while there was no significant difference among the applications and TA values changed between 5.61 g L^−1^ and 6.74 g L^−1^ in the second year [36].

The application of 10 μmol L^−1^ of BR on the eight-year-old “Shine Muscat” after 12 and 48 h increased the total organic acid content (13.47% and 14.3%, respectively) as compared to the control group (13.33% and 14.10%, respectively) during the fruit maturity stage [13]. Researchers used EBL as an exogenous application on twelve-year-old “Thompson seedless” table grapes and found that EBL treatments of 3 μmol L^−1^ and 6 μmol L^−1^ increased total organic acid content with the value of 0.675 g 100 g^−1^ and 0.670 g 100 g^−1^, respectively, but 3 μmol L^−1^ was significantly more effective [51]. Moreover, 100 μmol L^−1^ of BRs applied at the commencement of veraison on five-year-old “Kyoho” grapes had little influence on the content of organic acids. After 20 DAT (days after treatments), tartaric acid concentrations dropped from 80 mg g^−1^ to 70 mg g^−1^ and remained low during postharvest storage (60 mg g^−1^). Meanwhile, the content of organic acids in grape pericarp was higher (an average of 90 mg g^−1^) than in berry flesh (an average of 70 mg g^−1^), and tartaric acid accounted for a leading organic acid (more than 50%) [12]. Moreover, in one-year-old “Cabernet Sauvignon” grape seedlings, the application of 0.1 mg L^−1^ exogenous EBL significantly increased ascorbic acid (AsA) after 24 h of treatment (~190 mg 100 g^−1^) and 48 h treatment (~200 mg 100 g^−1^) compared with the control group (~140 mg 100 g^−1^, ~150 mg 100 g^−1^, respectively) [52].

As a result, we came to the conclusion that exogenous BR had a considerable impact on the organic acid in grape berries. However, studies at transcriptome and metabolomics levels are relatively rare and need to be implemented in future studies.

#### 3.2.4. Effect of BR on Astringency Reduction in Grape Berries

Tannins are key secondary metabolites in fruits, contributing significantly to astringency production. As the fruit ripens, the decrease process of the tannin substance begins, and then the substance is converted into water-soluble compounds, which improves the fruit’s taste [61]. Proanthocyanidins (PAs), also known as the most important condensed tannins, is a class of phenol that works to protect cytomembranes against free radicals, UV light, and oxidation [62]. Different PAs contents and kinds alter grape products taste, flavor, and sensory features due to their astringency and bitterness characteristics; their concentrations are also strongly affected by exogenous BR [63].

In particular, researchers used different concentrations of EBR (0.25 mg L^−1^, 0.5 mg L^−1^, 0.75 mg L^−1^, 1 mg L^−1^) solution to treat grape berries (*Vitis vinifera* cv. Cinsaul), finding that 0.5 mg L^−1^ and 0.75 mg L^−1^ of EBR significantly increased total tannins content (37.54 mg g^−1^ and 46.66 mg g^−1^, respectively) compared to the control group (29.76 mg g^−1^) [53]. Moreover, in a previous study, five-year-old “Cabernet Sauvignon” grape berries were treated with exogenous EBL. The 0.4 mg L^−1^ EBL treatments (twice at fruit set stage) were more effective for the production of total tannin (~95 mg g^−1^) in 60 DAA, following 0.4 mg L^−1^ EBL application (once at fruit set stage) and control group (~70 mg g^−1^) [44].

Researchers used different concentrations of EBL (0.1 mg L^−1^, 0.4 mg L^−1^, and 0.8 mg L^−1^) on “Yan 73” and “Cabernet Sauvignon” grape berries, and found that 0.4 mg L^−1^ of EBR treatment could promote the production of diphenyl picryl phenylhydrazine (DPPH), 2,2′-Azino-bis(3-ethylbenzothiazoline-6-sulfonic acid (ABTS), and hydroxyl radical-scavenging activity (HRSA), which were secondary metabolites and antioxidant parameters in these two grape varieties, as well as an increase of anthocyanin-related parameters such as total anthocyanin content (TAC), total flavonoids content (TFOC), total phenolics content (TPC), and total tannin content (TTC) by 16.0%, 8.2%, 40.0%, and 18.2%, respectively, in “Cabernet Sauvignon” and 28.0%, 9.4%, 19.4%, and 21.9%, respectively, in “Yan 73” grape berry [46]. Additionally, the synergistic effect of BL and light, which promotes anthocyanins accumulation in seven-year-old “Cabernet Sauvignon” grape pericarp, was discussed. The highest total anthocyanin contents were observed in 0.4 mg L^−1^ EBL + light condition treatment (0.862 mg g^−1^) following 0.4 mg L^−1^ EBL with dark condition (0.024 mg g^−1^) and control group (dark condition; 0.0089 mg g^−1^). Regarding anthocyanin monomers, 0.4 mg L^−1^ EBL with light condition treatment significantly increased the content of malvidin-3-O-glucoside (11.55 mg g^−1^; the most important substance in the anthocyanin monomer). Furthermore, in light conditions, other anthocyanin monomers such as malvidin-3-O-(6-O-acetyl)-glucoside, Ma-derivatives, and De-derivatives also showed the same increasing trend under 0.4 mg L^−1^ EBL treatment (4.5 mg g^−1^, 1.3 mg g^−1^, 0.25 mg g^−1^) compared with the dark condition treatment (1.5 mg g^−1^, 0.2 mg g^−1^, 0.01 mg g^−1^) [43]. In addition, through investigating the effects of three concentrations of EBR (0.1 mg L^−1^, 0.4 mg L^−1^, and 0.8 mg L^−1^) on six-year-old “Cabernet Sauvignon” grape berries, findings found that 0.4 mg L^−1^ EBR was the most optimal exogenous BR concentration, responsible for increasing total anthocyanins production in grape pericarp at a late fruit ripening stage (3.913 mg g^−1^) compared to the control group (3.369 mg g^−1^) [54]. Therefore, BR made a significant contribution to improving berry flavor, but further research is required to fully understand the mechanism by which BR reduces the astringency of grape berries.

### 3.3. Effect of BR on Grape Aroma Enrichment

#### 3.3.1. Aroma Components Existed in Grape Berries

Aromatic compounds such as esters, aldehydes, ketones, alcohols, acids, and terpenes exist in varying proportions in grapevines and play an important role in flavor formation and fruit quality. Additionally, the aroma characteristics of different grape varieties are formed by the interactions of these aromatic compounds, which have accumulative, synergistic, and inhibitory effects [64]. According to the previous study [65], amino acids, fatty acids, monosaccharides, glycosides, terpenes, hydroxyl acids, and phenols are precursors of aromatic compounds, which are formed through degradation and biosynthesis in the grape growth process under the activity of various enzymes. Among these, terpenes are the major contributor, even though they have a low concentration in grape berries, and still significantly contribute to the aroma formation in both grape berries and wine due to their low threshold value. As for table grapes, linalool was characterized as a large amount in *Muscat Hamburger*, while β-myrcene, trans-β-ocimene, α-ocimene, nerol, geraniol, and citral were detected in “Centennial Seedless”. Other terpenes compounds such as α-copaene and α-phellandrene were found in “Rizamat” and “Manaizi” grapes, whereas phytol and α-terpineol accounted for a significant proportion in “Red Globe” grapes [66]. As for wine, we considered the bound-form compounds (linalool glycoside is the most abundant terpene compound, followed by hotrienol and geranylacetone glycosides). In the case of a free-form compound, linalool was found with a higher concentration in Muscat wine. Furthermore, the significant contents of rose oxide are available as a bound-form compound, whereas geraniol and nerol (free-form) were in “Meili 3” wine [67].

#### 3.3.2. The Effect of BR on Aroma Components in Grape Berries

The regulating effect of BR on terpene biosynthesis has been reported in different studies [68] (Table 3). For example, a report has verified that 100 μmol L^−1^ of exogenous BRs could change the aroma composition of five-year-old “Kyoho” grapes, and increase the proportion of terpenes compounds such as α-pinene (0.03%), d-limonene (0.07%), and γ-terpene (63.67%) compared with the control group (0.01%, 0.04%, and 62.96%, respectively). The application of 100 μmol L^−1^ BR is responsible for enhancing the concentration of terpene compounds because it has the capability to trigger the regulation of the terpene biosynthesis pathway [12]. Moreover, after the application of 10 μmol L^−1^ BR on eight-year-old “Shine Muscat” grape berries after 48 h, the content of linalool (16.29%) was significantly higher than the control group (12.68%). Furthermore, geraniol only existed in those grapes that were treated with the combined application of 10 μmol L^−1^ ABA + 10 μmol L^−1^ BR (0.56%) [13].

In addition, aromatic compounds such as alcohols and aldehydes also contributed greatly to the aroma components of grape berries. According to current research [13], 10 μmol L^−1^ exogenous BR plays an essential role in boosting the concentration of alcohols and aldehydes in eight-year-old “Shine Muscat” grape berries, as evidenced by a two-fold rise of 1-pentanol and sinapyl alcohol proportions than the control group after 48 h treatment. Additionally, alcoholic compounds such as 1-hexanol and acetoin proportion (4.68% and 10.35%, respectively) were significantly more in content than the control group (2.92%, 5.73%, respectively). In the case of aldehydes, the proportion of methyl glyoxal (12.12%), pentanal (2.31%) as well as hexanal (22.96%) were found in abundance as compared with the control group (0%, 2.08%, and 15.27%, respectively). At present, there are more pieces of evidence that exogenous BR is closely related to the biosynthesis of free-form terpenes in grape berries, but fewer reports on exogenous BR influence the biosynthesis of glycosidic bound-form terpenes. It will have broad prospects to investigate the benefits of exogenous BR on the accumulation of bound-form terpenes.

#### 3.3.3. Regulating the Effect of BR on Key Rate-Limiting Enzyme HMGR

Isoprenoid substances are mainly biosynthesized by a series of enzymes in the mevalonate metabolic pathway, in which 3-Hydroxy-3-methylglutaryl CoA reductase (HMGR) serves as the first major rate-limiting enzyme in this pathway [69]. HMGR not only converted 3-hydroxy-3-methylglutaryl coenzyme A to mevalonate, but it also played a vital role in terpenes metabolism [70]. The *HMGR* family in the grapevine has three members, *VvHMGR1*, *VvHMGR2*, and *VvHMGR3*, and their expression in the “Kyoho” grape showed obvious tissue specificity, in which *VvHMGR1* was highly expressed, and its expression level in leaves, flowers, and pericarp was much higher than that in other organs. Moreover, the expression level of VvHGMRs in yellow–green varieties was significantly higher than that in red varieties. In the case of rose-scented varieties, the expression level was 1.20–1.25 folds higher than that in strawberry-scented varieties [71]. There was some research about the regulation of exogenous BR on HMGR activity, validating that its expression treated by 100 μmol L^−1^ of exogenous BR was significantly higher (10 times up-regulation) than the control group at the beginning and half veraison stages of five-year-old “Kyoho” grapevine [12]. Moreover, the BR biosynthetic gene *cytochrome P450 steroid monooxygenase B3* (*SlCYP90B3*) in tomatoes was detected via an ethylene-dependent pathway. Overexpression of such genes could activate the accumulation of carotenoids and the formation of phenols substances, which influences the aroma component [7]. Regarding *Arabidopsis thaliana*, a lower content of campestanol, campesterol, and sterol in *hmg1-1* seedlings (39%, 75%, and 47%, respectively) and inflorescence (75%, 85%, and 25%, respectively) was found compared to that of WT (wide type) seedlings, but the cholesterol level in *hmg1-1* and WT was at par [72]. Although no evidence has been found that exogenous BR has the greatest influence on which structural genes or transcription factors, we still conclude that BR was closely related to terpene biosynthesis. However, the specific molecular mechanism remained unclear and needed further verification.

### 3.4. Effect of BR on Grape Pericarp Coloration

Grapes have a large market potential due to their unique physical color attraction to consumers. Additionally, consumers choose grape varieties based on the color of the pericarp, and therefore producers want to have superior color-quality grapes [73]. A key flavonoid substance called “Anthocyanin” influences the color of grapes and is responsible for the high quality of grape pericarp and wine [74]. Previous research has shown that a variety of factors influence anthocyanin biosynthesis, including temperature, light, water, soil, climate, etc. For example, high temperatures can inhibit anthocyanin accumulation, whereas strong light, adequate water, and proper nutrition can promote pericarp coloration [17,75,76,77]. Furthermore, an improved anthocyanin extraction procedure could lead to higher anthocyanin content in grapes and good wine quality [78].

Researchers sprayed 0.06 mg L^−1^ BR exogenously on six-year-old “Red Globe” and ten-year-old “Crimson seedless” grapevines at the start of the version (B60) and again seven days later (B60 + B60). They found that treatment B60 and B60 + B60 had significantly greater anthocyanin concentrations in “Red Globe” (~1.45 mg berry^−1^ and ~1.3 mg berry^−1^) and “Crimson Seedless” (~3.5 mg berry^−1^ and ~4.5 mg berry^−1^) grape berries, respectively, in comparison to control (~1.2 and ~3 mg berry^−1^, respectively) [55]. In addition, after applying 0.1, 0.4, and 0.8 mg L^−1^ exogenous EBR to a six-year-old grapevine, researchers discovered that 0.4 mg L^−1^ exogenous EBR was the most effective treatment for increasing total anthocyanin content, and significant color development was observed 7 days earlier than the control group [54]. Additionally, the same concentration of BR (0.4 mg L^−1^) was proven as the best treatment against anthocyanin compounds of sixteen-year-old self-rooted “Redglobe” grapevine. The content of cyanidin-3-glucoside and delphinidin-3-glucoside (anthocyanin compounds) in the grape pericarp was ~1.5 times more than the control group [41]. In addition, according to a previous study, they sprayed 100 M BRs on five-year-old “Kyoho” grapes and reported that it was more effective for higher anthocyanin concentration and contents were higher at the outset of version 0.0998 mg g^−1^, followed by half-version (0.0560 mg g^−1^), and full-version (0.0281 mg g^−1^). During postharvest storage, the anthocyanin content of 100 μmol L^−1^ BR-treated grapes was higher than that of control-treated (value) [12]. Applying 0.6 mg L^−1^ BR to nine-year-old Alphonse Lavallée (*Vitis vinifera* L. cv.) grapevines three times (7 days after berry set + veraison + 30 days after version) produced the maximum anthocyanin content in both growing years (75.89 mg 100 g^−1^ and 86.90 mg 100 g^−1^, respectively), compared to the control group (42.82 mg 100 g^−1^ and 49.53 mg 100 g^−1^, respectively) [36].

A previous study investigated the influence of BR on the five-year-old “Yan 73” grapevine during the ripening period to better understand anthocyanin-based enzymatic behavior and found that the activity of phenylalanine ammonia-lyase (PAL) gradually rose, whereas the activity of UDP-glycose flavonoid glycosyltransferase (UFGT) increased and subsequently dropped. When compared to the control group, a concentration of 0.8 mg L^−1^ BR significantly improved the activity of PAL (130 U g^−1^) and UFGT (28 U g^−1^) during the late stage of fruit ripening [79]. Additionally, a study on five-year-old “Cabernet Sauvignon” grapes used varied concentrations of EBL (0.1 mg L^−1^, 0.4 mg L^−1^, and 0.8 mg L^−1^). They concluded that 0.4 mg L^−1^ EBL significantly up-regulated the activities of key rate-limiting enzymes such as UFGT (~9 mmol g^−1^) and PAL (~240 U g^−1^) in comparison with the control group (~5 mmol g^−1^ and ~210 U g^−1^, respectively). The same results were achieved after the application of 0.4 mg L^−1^ EBL to a five-year-old “Yan 73” grapevine, which had considerably higher UFGT (~23 mmol g^−1^) and PAL (~170 U g^−1^) activity than the control group (~18 mmol g^−1^ and ~150 U g^−1^, respectively) [46].

Besides that, exogenous BR contributing to increasing grape pericarp coloration at the molecular level was widely discussed. Overall, 0.4 mg L^−1^ exogenous EBL promoted the accumulation of total anthocyanins (~5 mg g^−1^) in seven-year-old own-rooted “Cabernet Sauvignon” grape berries compared to the control group (~2.5 mg L^−1^), and developed its pericarp color after 46 days treatment by up-regulating the expressions of genes *chalcone isomerase1* (*VvCHI1*), *chalcone synthase3* (*VvCHS3*), *flavonoid-3′, 5′-hydroxylase* (*VvF3′5′H*), *dihydroflavonol reductase* (*VvDFR*), *VvUFGT* (about 1.2–3 times increment). Exogenous EBL application during grape berry development on seven-year-old “Cabernet Sauvignon” influenced the expression profiles of transcription factors MYB proteins (MYB5a, MYB5b, and MYBPA1) interacting with anthocyanin biosynthesis structural genes, as seen in the late stage of grape berry ripening (120 days after anthesis). The EBL treatment (0.4 mg L^−1^) dramatically boosts the relative expressions of MYB5a (~0.5), MYB5b (~0.25), and MYBPA1 (~7) at 46 days after treatment as compared to the control (~0.2, ~0.15, ~2, respectively) [43]. Light could interact with BR signals, jointly regulating the coloration of the grape pericarp, and transcription factors of light signals like ELONGATED HYPOCOTYL5 (HY5), GATA binding protein 2 (GATA2), and Constitutively Photomorphogenic 1 (COP1) could interact with transcription factors like BZR1 and BIN2 in the BR signaling pathway, where these transcription factors with photoreactive activity combine in the form of dephosphorylation and jointly regulate the transcription of the key gene UFGT related to anthocyanins [80].

The findings strongly suggested that exogenous BR not only promoted anthocyanin accumulation in the grape pericarp but also regulated the expression of genes involved in anthocyanin production and metabolism, however, the exact molecular mechanism is yet unknown. Table 4 highly generalized some findings regarding the involvement of genes in grapes quality with respect to BR. These genes may be extremely important for the study of the signal interaction between BR signal and quality-related metabolites, which needs extensive investigation.

### 3.5. Effect of BR on Grape Antioxidant Metabolites Accumulation

Plants have developed an antioxidant system to minimize the formation of reactive oxygen species, which were in a dynamic equilibrium state under normal plant development conditions [81]. Grapevines have been shown to respond to drought by regulating many secondary metabolic pathways, mainly by stimulating the synthesis of phenylpropanoid, carotenoid zeaxanthin, and volatile organic compounds, which affects the antioxidant potential, composition, and sensory characteristics of grape berries and wine [82].

In addition, it is also frequently reported that BR regulates the antioxidant activity of grapes. For example, 0.1 mg L^−1^ exogenous application of EBL improved the ability of one-year-old “Cabernet Sauvignon” grape seedlings to resist low-temperature stress, as evidenced by a significant decrease in hydrogen peroxide (H_2_O_2_, 9 mol g^−1^) and reactive oxygen (ROS, 14 g g^−1^) after 6 h EBL treatment compared to the control group. After 12 h of treatment, there was a substantial increase in superoxide dismutase (SOD, 13 U g^−1^ min^−1^) compared to the control group (9 U g^−1^ min^−1^) [52]. Moreover, researchers used two concentrations of BR (0.75 ppm and 1.5 ppm) on one-year-old “Rish Baba” grapes, and the results indicated that 1.5 ppm application significantly increased the levels of catalase (CAT, ~55 U mg^−1^) and peroxidase (POD, ~40 U mg^−1^), while 0.75 ppm treatment was responsible for enhancing the activity of ascorbate peroxidase (APX, ~35 U mg^−1^) rather than the control group [37]. After spraying EBL on the twelve-year-old “Thompson Seedless” grape, both EBL applications (3 μmol L^−1^ and 6 μmol L^−1^) significantly promote the activities of polyphenol oxidase with the value of ~12,000 U mg^−1^ and ~13,000 U mg^−1^, respectively [51]. Interestingly, the content of total antioxidant activity (TAA) was significantly increased under 3 μmol L^−1^ EBR treatment (~4000 mmol 100 g^−1^) compared with the control group (~2000 mmol 100 g^−1^). Besides, by using EBR on “Cabernet Sauvignon” grape seedlings under drought stress, findings discovered that 0.1 mol L^−1^ EBR significantly enhanced the level of APX activity after 12 h (~1.1 U g^−1^) and 24 h (~1.15 U g^−1^) of treatment as compared to control (stressed and unstressed). Similar to the variation tendency of APX activity, the EBR application was also responsible to enhance the considerable activity of glutathione reductase (GR) at 48 h and 72 h of treatment (~0.045 U g^−1^ and ~0.04 U g^−1^, respectively) compared with the drought stress treatment group (~0.03 U g^−1^ and ~0.035 U g^−1^, respectively) [47].

The results demonstrated a synergistic relationship between the BR signal and a rise in grapevine antioxidant activity, although the particular regulatory mechanism is still unknown. The findings addressing the role of enzymes in grape quality in relation to BR are summarized in Table 5, as they serve as guidelines for studying the intrinsic mechanism of BR on these enzymes’ activity.

## 4. The Interaction of Endogenous Phytohormones Regulated Plant Growth and Development

Endogenous phytohormones were found to have a strong relationship with plant growth activities such as leaf senescence, flower development and senescence, fruit ripening, fruit quality improvement, and stress tolerance improvement [84,85,86]. Previous research has found that BR has an antagonistic effect on numerous endogenous phytohormones (ABA/IAA/SA/gibberellin/Jasmonic acid (JA)/ETH) that affect fruit quality and stress resistance in *Arabidopsis thaliana* (The structures of all the mentioned phytohormones were shown in the Figure 4). The interaction between brassinosteroid insensitive receptor (BIN2) and ABSCISIC ACID INSENSITIVE5 (ABI5) inhibits the germination and blooming phase of seeds because exogenous BR can reduce ABA signaling and is responsible for triggering the stomatal opening in leaves [87]. Meanwhile, when the mutant aba1-1 (ABA-deficient) was exposed to heat stress, it increased the accumulation of heat shock protein (HSP) and BR production, implying that the antagonistic effect between BR and ABA was crucial for improving heat tolerance in *Arabidopsis thaliana* [88]. In terms of BR-IAA interaction, previous research has shown that BR interferes with IAA to improve heat stress resistance in *Arabidopsis thaliana* by repressing transcription factors (IAA/AUX1, PIN4) in the IAA biosynthesis pathway. Additionally, IAA and BR interactions may occur at promoters of common target genes (*DET2*, *DWF4*), influencing their transcriptions [89]. Besides, overexpression of the BR signal element SlBZR1 improved the cold tolerance of tomato fruit by upregulating the expression of genes related to low-temperature stress (*SlICE1*, *SlCBF1*, *SlCBF2*, and *SlCBF3*), as well as a significantly higher IAA content in SlBZR1 overexpression plants (32 ng g^−1^) than in WT plants (25 ng g^−1^) after low-temperature stress [90].

There were also some studies that looked at how BR and other hormones like SA/gibberellin/JA/ETH interacted to improve plant stress and disease resistance. Both exogenous and endogenous applications of BRs could negatively affect the immunity of pathogen *P. graminicola* in rice roots, and this immunosuppression was illustrated to have at least partial crosstalk with SA and gibberellin pathways. BR not only antagonized SA-mediated immune defense but also affected the expression of non-expression factor 1 (OsNPR1) and OsWRKY45 (vital transcription factors in SA response). In addition, by interfering with gibberellin metabolism at several levels, BR suppresses gibberellin-directed immunological responses, therefore stabilizing the DELLA protein and the key gibberellin repressor SLENDER RICE1 (SLR1) [91]. The non-expression factor 1 (NPR1) of pathogenic-associated genes was found to be a critical component of BR-mediated effects on heat and salt tolerance in Arabidopsis, and ABA inhibited the BR effect during heat stress. Additionally, BR was found to share transcriptional targets with other phytohormones (ABA, SA, and JA) in Arabidopsis, implying that BR will interact with a variety of endogenous phytohormones. In addition to NPR1, WRKY70 could be another potential crosstalk link between SA and BR, through which BR could stimulate some SA response genes (*PR-1*, *GST1*), and the overall induction of WRKY70 by SA would also need the presence of a functional NPR1. When 1 M of BR was applied to a JA mutant, the expression of the JA response gene (PDF1.2) increased significantly compared to the control group, demonstrating that there was a synergy between BR and JA as well [88]. Furthermore, ETH and BR interactions influenced the expression of IAA signaling transcription factors (ERF1, ERF5) to improve plant resistance against Botrytis cinerea [92].

According to the findings, most studies were performed on the model plant to understand the interaction of BR and other phytohormones. In this study, phenotypic changes in grape or tomato fruit caused by the interaction of BR and ABA/ETH were summarized and explained in a model diagram (Figure 5). In the future, research is needed to understand how phytohormones interact with BR to modulate grapes quality.

## 5. Conclusions

In this review, the particular association between BR and grape quality improvement was explored based on prior development in BR biosynthesis and metabolism, BR signal transduction, grape external quality enhancement, and biochemical property generation by the application of exogenous BR. Overall, we found that BR has a clear regulatory effect on fruit quality (external quality and internal quality), although the exact molecular mechanism is still being investigated. Further research is needed to understand how the interaction of BR with other endogenous phytohormones affects grape berry development and ripening, as well as the involvement of molecular mechanisms. Furthermore, it is necessary to clarify the role of BR in grapevine development and provide guidance for grape cultivators who use exogenous BR spraying in grapevine cultivation.

## Figures and Tables

**Figure 1 ijms-24-00445-f001:**
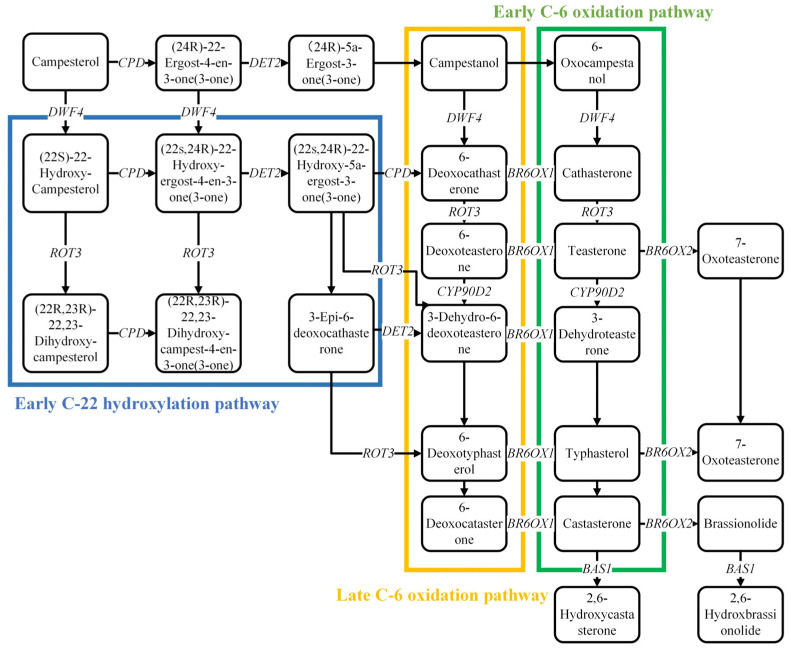
The biosynthesis and metabolism pathways of BR. The blue frame expressed the early C-22 hydroxylation pathway in BR biosynthesis, the yellow frame expressed the late C-6 oxidation pathway, and the green frame expressed the early C-6 oxidation pathway. There are six rate-limiting enzymes of BR biosynthesis, namely *CPD*, *DWF4*, *ROT3*, *DET2*, *BR6OX1*, *BR6OX2*; and one rate-ling enzyme of BR metabolism, namely *BAS1*.

**Figure 2 ijms-24-00445-f002:**
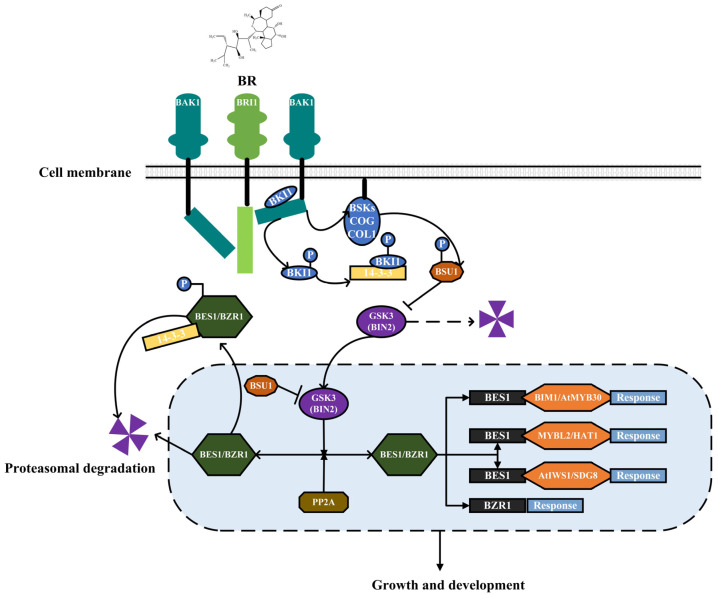
The signal transduction of BR. The cylinder (BAK1; BRI1), the circle (BKI1; BSKs/COG/COL1; GSK3; BIN2), lozenge (BSU1; BES1/BZR1; PP2A; BIM1/AtMYB30; MYBL2/HAT1; AtIWS1/SDG8), and rectangle (14-3-3; BES1) denoted the receptor proteins or signal molecules involved in signal transduction. (→) indicated the promoting effect, (⟞) indicated the inhibiting effect.

**Figure 3 ijms-24-00445-f003:**
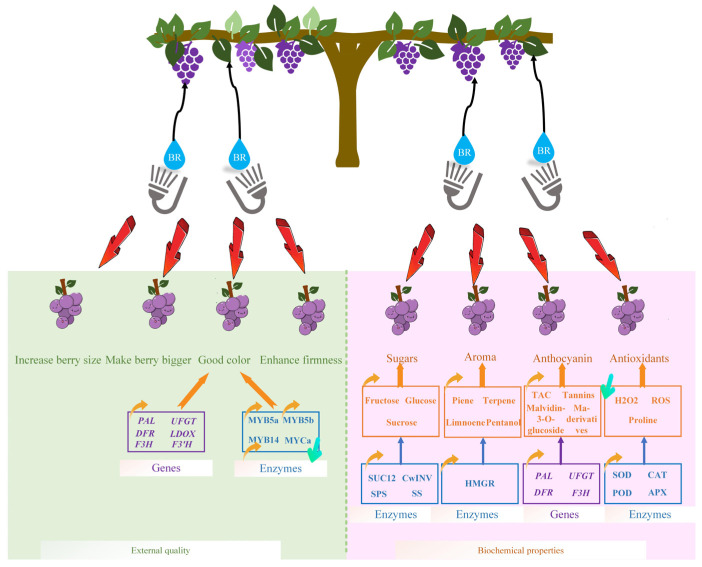
The effect of exogenous BR on the berry external quality improvement and biochemistry biochemical properties formation. External quality included berry size, berry color, and berry firmness. Biochemical properties include sugars (fructose, glucose, and sucrose), aroma substances (β-pinene, terpene, limonene, pentanol), anthocyanin (TAC, tannins, malvidin-3-O-glucoside, Ma-derivatives), and antioxidants (SOD, CAT, POD, APX). The mentioned enzymes and genes related to grape berry quality improvement are also listed in the figure.

**Figure 4 ijms-24-00445-f004:**
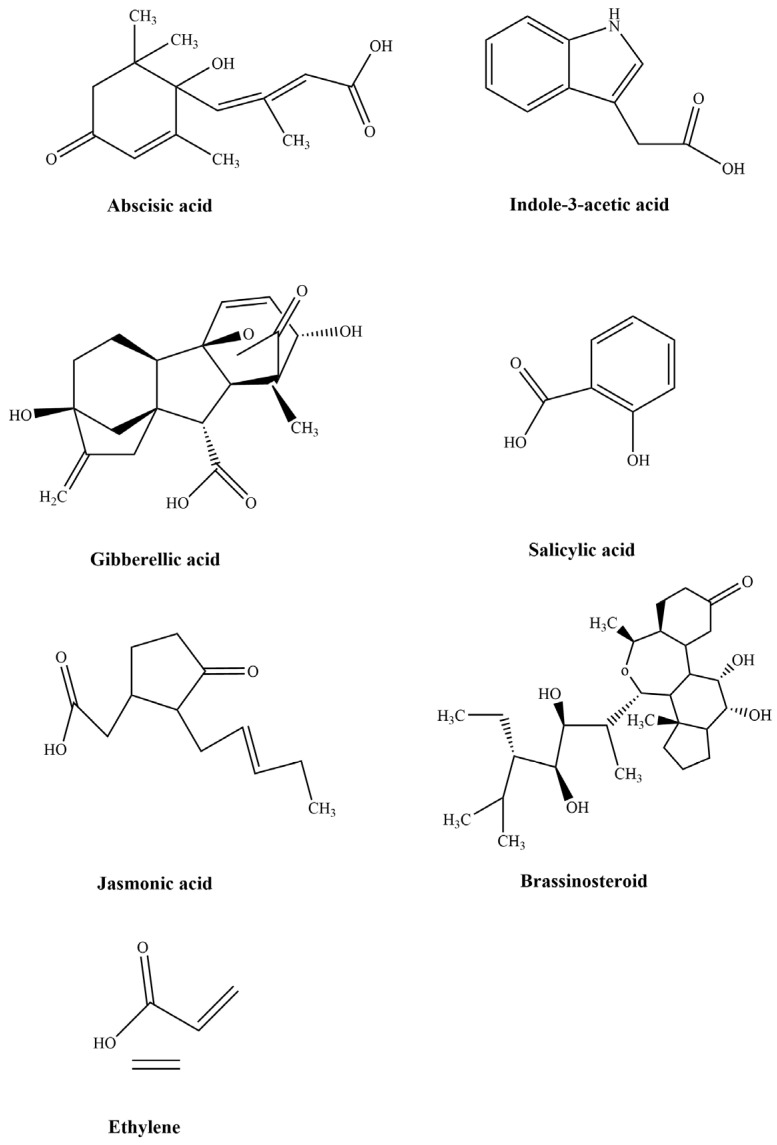
The molecular structures of all the mentioned phytohormones, including abscisic acid, Indole-3-acetic acid, gibberellic acid, salicylic acid, jasmonic acid, brassinosteroid, and ethylene. The figure was drawn by ChemDraw 2022 (CambridgeSoft, Cambridge, MA, USA).

**Figure 5 ijms-24-00445-f005:**
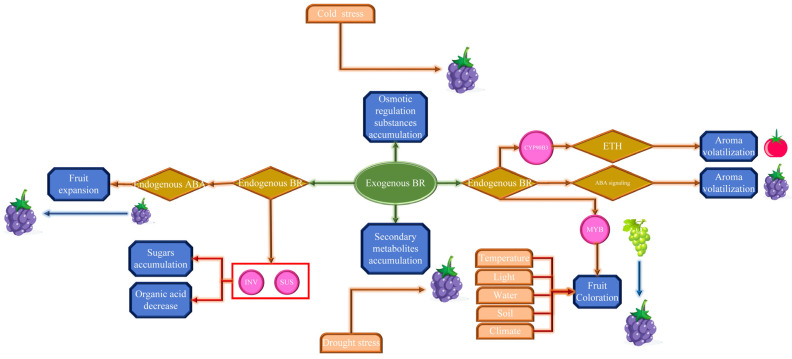
The effect of BR on the berry development and fruit quality improvement. Dark-blue frame displayed the main physiological activity related to fruit quality improvement, including fruit expansion, sugars accumulation, organic acid decrease, fruit coloration and aroma volatilization. The tan frame and green frame display various endogenous phytohormones, such as ABA, BR, and ETH. Incarnadine frame displayed different environmental factors, namely temperature, light, water, soil, climate, and stress. The pink frame displays structural genes and transcription factors involved in these processes, comprising *INV*, *SUS*, MYB, and CYP90B3. (→) indicated the regulating effect, while (⟞) indicated inhibiting effect.

**Table 1 ijms-24-00445-t001:** Role of exogenous brassinosteroid/brassinolide on the growth or development of external grape berries quality.

S. No	Compounds	Best Concentration (From Literature Sources)	Best Concentration (Concerted to Molar)	Variety	Age of Grapevine	Key Findings
1	Brassinosteroid	100 μmol L^−1^	100 μmol L^−1^	Kyoho	5-year	BR treatment improved grape pericarp coloring at various phases of fruit development, which had most noticeable effect occurring at the start of veraison [12].
2	Brassinosteroid	0.2 mg L^−1^	0.42 μmol L^−1^	Alphonse Lavallée	9-year	Responsible for the maximum berry weight (7.65 g and 7.87 g), cluster weight (374.98 g and 418.75 g), and yield production (26.25 vine kg^−1^ and 30.15 vine kg^−1^) in both production years, respectively [36].
3	Brassinosteroid	1.5 ppm	4.58 μmol L^−1^	Rish Baba	Not mentioned	1.5 ppm BR treatment was effective and responsible for reducing single fruit weight loss from 29.75% to 29.48% under cold storage for 5 weeks [37].
4	Brassinosteroid	0.5 ppm and 1 ppm	1.53 μmol L^−1^ and 3.05 μmol L^−1^	Flame Seedless	12-year	0.5 ppm and 1 ppm of BR significantly increased berry weight (2.55 and 2.57 g), berry length (1.84 and 1.89 cm), and berry breadth (1.73 and 1.74 cm) in comparison to the control group (2.38, 1.71, and 1.59 cm, respectively) [38].
5	Brassinosteroid	0.5 ppm and 1 ppm	1.53 μmol L^−1^ and 3.05 μmol L^−1^	Flame Seedless	12-year	The application of BRs on the grape cluster effectively delayed the deviation rates of *L**, *A**, and *B** and was responsible for color changing from relatively pure green to yellow and subsequently to red [38].
6	24-Epibrassinosteroid	0.4 mg L^−1^	0.83 μmol L^−1^	Khalili	8-year	0.4 mg L^−1^ EBR significantly increased the cluster weight (223.77 g), which was higher than that of 0.2 mg L^−1^ EBR treatment (208.30 g) and the control group (185.69 g). As well, the cluster length of 0.4 mg L^−1^ (25.26 cm) was higher than that of 0.2 mg L^−1^ EBR treatment (23.16 cm) and the control group (19.29 cm) [42].
7	24-Epibrassinolide	0.4 mg L^−1^	0.83 μmol L^−1^	Cabernet Sauvignon	7-year	Exogenous EBL promoted the total weight of 100-berry of Cabernet Sauvignon (~140 g) as compared to the control group (~135 g) after 46 days of treatment [43].
8	24-Epibrassinolide	0.4 mg L^−1^	0.83 μmol L^−1^	Cabernet Sauvignon	5-year	The EBL treatment (twice at fruit setting), and EBL application (once at fruit set stage), were effective for the development of cluster weights (190 g and 170 g, respectively) in 116 DAA, in comparison with the control group (140 g) [44].
9	Brassinolide	0.5 ppm	1.53 μmol L^−1^	Thompson seedless grapes	4-year	BL could significantly improve the cluster weight and yield of grape berries, which was 1–2 times higher than that of the control group [45].
10	24-Epibrassinolide	0.4 mg L^−1^	0.83 μmol L^−1^	Cabernet Sauvignon	Not mentioned	The single berry weight (~1.3 g) of grapes sprayed with EBL was significantly higher than that of the control group (~1.2 g) [46].
11	24-Epibrassinolide	0.4 mg L^−1^	0.83 μmol L^−1^	Yan 73	Not mentioned	The single berry weight (~1.5 g) of grapes sprayed with EBL was significantly higher than that of the control group (~1.6 g) [46].
12	ABA, Brassinosteroid, ABA + Brassinosteroid	10 μmol L^−1^	10 μmol L^−1^	Shine Muscat	8-year	At 12, 24, 36, and 48 h, BR treatment gained the weight (69.39, 69.18, 68.99, and 68.89, respectively), followed by BR + ABA (65.87, 65.65, 65.46, and 65.39, respectively), control (65.77, 65.55, 65.30, and 65.23, respectively), and ABA (65.74, 65.53, 65.33, and 65.26, respectively) [13].
13	28-homobrassinolide	8 ppm	16.19 μmol L^−1^	Sultani	12-year	Exogenous application of 28-homobrassinolide (8 ppm) improved fruit firmness (7.11 N) more in grape berries than the control group (6.19 N) [39].
14	24-Epibrassinolide	0.8 mg L^−1^	1.66 μmol L^−1^	Red globe	5-year	The fruit firmness was decreased after 15 days (5.25 N), 30 days (5.15 N), and 60 days (4.98 N) of storage under 0.8 mg L^−1^ EBL treatment [40].
15	3α-hydroxy-20-RB-homo-7-oxa-5α-cholestan-6-one	0.4 mg L^−1^	0.83 μmol L^−1^	Red globe	16-year	Exogenous BR significantly contributed to the CIRG (the color parameter of red grape variety, which was evaluated according to CIELAB parameters *L** (brightness), *H* (tone angle), and *C* (chroma) of sixteen-year-old self-rooted “Redglobe” grapevine [41].
16	24-Epibrassinosteroid	0.1 μmol L^−1^	0.1 μmol L^−1^	Cabernet Sauvignon grape seedlings	Not mentioned	EBR treatment was more effective in alleviating the damage of grapevine phenotypes under water stress. Compared with the control group, EBR had less damage (moderate dehydration, drooping and curling of leaves) [47].

**Table 2 ijms-24-00445-t002:** Effect of exogenous brassinosteroid/brassinolide on biochemical properties of grapes.

S. No	Compounds	Best Concentration (From Literature Sources)	Best Concentration (Concerted to Molar)	Variety	Age of Grapevine	Key Findings
1	Brassinosteroid	100 μmol L^−1^	100 μmol L^−1^	Kyoho	5-year	BRs applied at the commencement of veraison on grapes had little influence on the content of organic acids. After 20 DAT (days after treatments), tartaric acid concentrations dropped from 80 mg g^−1^ to 70 mg g^−1^ and remained low during postorbital storage (60 mg g^−1^). Meanwhile, the content of organic acids in grape pericarp was higher (an average of 90 mg g^−1^) than in berry flesh (an average of 70 mg g^−1^), and tartaric acid accounted for a leading organic acid (more than 50%) [12].
2	Brassinosteroid	100 μmol L^−1^	100 μmol L^−1^	Kyoho	5-year	At 30 days after BR treatment, anthocyanin content was 0.0998 mg g^−1^ in the BR treatment group, the value was higher than in control groups of the initiation of the veraison, half veraison stage (0.0560 mg g^−1^), and the full veraison stage (0.0281 mg g^−1^). During postharvest storage, the anthocyanin content of BR-treated grapes was higher than that of control-treated grapes [12].
3	Brassinosteroid	0.6 mg L^−1^	1.25 μmol L^−1^	Alphonse Lavallée	9-year	In both growing years, compared with the control group (42.82 mg 100 g^−1^, 49.53 mg 100 g^−1^) and other BR concentrations, the application of 0.6 mg L^−1^ of BR to vines three times (7 days after berry set + veraison + 30 days after veraison) furnished the maximum anthocyanin content (75.89 mg 100 g^−1^ and 86.90 mg 100 g^−1^) [36].
4	24-Epibrassinosteroid	0.4 mg L^−1^	0.83 μmol L^−1^	Khalili	8-year	EBR significantly increased the total soluble solid content (22.26 °Bx), which was higher than that of 0.2 mg L^−1^ EBR treatment (21.33 °Bx) and the control group (18.94 °Bx) [42].
5	24-Epibrassinolide	0.4 mg L^−1^	0.83 μmol L^−1^	Cabernet Sauvignon	7-year	The highest total anthocyanin contents were observed in 0.4 mg L^−1^ EBL + light condition treatment (0.862 mg g^−1^), following 0.4 mg L^−1^ EBL with dark condition (0.024 mg g^−1^) and control group (dark condition; 0.0089 mg g^−1^). The 0.4 mg L^−1^ EBL with light condition treatment (11.55 mg g^−1^) significantly increased the content of malvidin-3-O-glucoside [43].
6	24-Epibrassinolide	0.4 mg L^−1^	0.83 μmol L^−1^	Cabernet Sauvignon	5-year	The EBL treatments (twice at the fruit set stage) were more effective for the production of total tannin (95 mg g^−1^) in 60 DAA, followed by EBL application (once at fruit set stage) and the control group (70 mg g^−1^) [44].
7	24-Epibrassinolide	0.40 mg L^−1^	0.83 μmol L^−1^	Cabernet Sauvignon and Yan 73	Not mentioned	The total soluble solids (19 °Bx and 18 °Bx, respectively) or reducing sugar content (167 g L^−1^ and 165 g L^−1^, respectively) of grapes sprayed with EBL were significantly higher than that of the control group (Total soluble solids: 17 °Bx and 15 °Bx, respectively; reducing sugar: 160 g L^−1^ and 150 g L^−1^, respectively) [46].
8	24-Epibrassinolide	0.4 mg L^−1^	0.83 μmol L^−1^	Yan 73 and Cabernet Sauvignon	Not mentioned	EBL treatment could promote the production of DPPH, ABTS, and HRSA (secondary metabolites and antioxidant parameters) in these two grape varieties, as well as an increase of anthocyanin monomers such as TAC, TFOC, TPC, and TTC by 16.0%, 8.2%, 40.0%, and 18.2% respectively, in “Cabernet Sauvignon” grape berry and 28.0%, 9.4%, 19.4%, and 21.9%, respectively, in “Yan 73” grape berry [46].
9	ABA, Brassinosteroid, ABA + Brassinosteroid	10 μmol L^−1^	10 μmol L^−1^	Shine Muscat	8-year	The application of BR on the grapevine after 12 and 48 h were significantly increased the tartaric acid content (0.5 mg g^−1^ and 0.5 mg g^−1^, respectively) and total organic acid content (0.9 mg g^−1^ and 0.8 mg g^−1^, respectively) as compared to control group (tartaric acid content: 0.4 mg g^−1^ and 0.4 mg g^−1^, respectively; total organic acid content: 0.8 mg g^−1^ and 0.8 mg g^−1^, respectively) during the fruit maturity stage [13].
10	28-homobrassinolide	0.4 mg L^−1^	1.22 μmol L^−1^	Redglobe	16-year	The content of cyanidin-3-glucoside and peonidin-3-glucosidein (anthocyanin compounds) in the grape pericarp was ~1.5 times more than the control group under BL treatment [39].
11	24-Epibrassinolide	0.6 mg L^−1^	1.25 μmol L^−1^	Merlot	10-year	EBL treatment significantly enhanced the accumulation of monosaccharides in grape berries, including an increase in glucose and fructose content (16.95% and 39.31%, respectively) at the maturity stage, compared to the control [50].
12	24-Epibrassinolide	3 μmol L^−1^ and 6 μmol L^−1^	3 μmol L^−1^ and 6 μmol L^−1^	Thompson Seedless	12-year	3 and 6 μmol L^−1^ EBL significantly increased TSS levels (22 °Bx, 22.5 °Bx, respectively) as compared to the control group (18.5 °Bx) [51].
13	24-Epibrassinolide	3 μmol L^−1^ and 6 mol L^−1^	3 μmol L^−1^ and 6 μmol L^−1^	Thompson Seedless	12-year	3 and 6 μM EBL significantly increased TSS levels (22 °Bx, 22.5 °Bx, respectively) as compared to the control group (18.5 °Bx) [51].
14	24-Epibrassinolide	3 μmol L^−1^	3 μmol L^−1^	Thompson seedless	12-year	EBL treatments could significantly increase total organic acid content with the value of 0.675 g 100 g^−1^ and 0.670 g 100 g^−1^, respectively [51].
15	24-Epibrassinolide	0.1 mg L^−1^	0.31 μmol L^−1^	Cabernet Sauvignon	1-year	The application of exogenous EBL significantly increased ascorbic acid (AsA) after 24 h treatment (~190 mg 100 g^−1^) and 48 h treatment (~200 mg 100 g^−1^) compared with the control group (~140 mg 100 g^−1^, ~150 mg 100 g^−1^, respectively) [52].
16	24-Epibrassinolide	0.1 mg L^−1^	0.21 μmol L^−1^	Cabernet Sauvignon	1-year	Exogenous application of EBL improved the ability of one-year-old “Cabernet Sauvignon” grape seedlings to resist low-temperature stress, as evidenced by a significant decrease in H_2_O_2_ (9 mol g^−1^) and ROS (14 g g^−1^) after EBL 6 h treatment compared to the control group [52].
17	24-Epibrassinosteroid	0.5 mg L^−1^ and 0.75 mg L^−1^	1.04 μmol L^−1^ and 1.56 μmol L^−1^	Cinsaul	Not mentioned	0.5 mg L^−1^ and 0.75 mg L^−1^ of EBR significantly increased total tannins content (37.54 mg g^−1^ and 46.66 mg g^−1^, respectively) compared to the control group (29.76 mg g^−1^) [53].
18	24-Epibrassinosteroid	0.4 mg L^−1^	0.83 μmol L^−1^	Cabernet Sauvignon	6-year	Exogenous EBR was the most effective treatment for increasing total anthocyanin content and significant color development was observed 7 days earlier than in the control group [54].
19	24-Epibrassinosteroid	0.4 mg L^−1^	0.83 μmol L^−1^	Cabernet Sauvignon	6-year	EBR was responsible for increasing total anthocyanins production in grape pericarp at late fruit ripening stage (3.913 mg g^−1^) compared to the control group (3.369 mg g^−1^) [54].
20	BR, BR treated twice after the first BRs	0.06 mg L^−1^	0.18 μmol L^−1^	Red Globe and Crimson Seedless	6-year	They found that BR had significantly greater anthocyanin concentrations in Redglobe (~1.45 mg berry^−1^) and Crimson Seedless (~4.5 mg berry^−1^) grape berries, in comparison to control (~1.2 and ~3 mg berry^−1^, respectively) [55].
21	24-Epibrassinolide	0.42 μmol L^−1^ and 0.21 μmol L^−1^	0.42 μmol L^−1^ and 0.21 μmol L^−1^	Cabernet Sauvignon	1-year	On the 3rd–5th day after EBL treatment, the total protein content of EBL1 (0.21 μM) and EBL2 (0.42 μM) treatment increased by 12% and 13%, respectively, compared with the control [56].

**Table 3 ijms-24-00445-t003:** Impact of exogenous brassinosteroid/brassinolide on the production of aroma in different grape varieties.

S. No	Compounds	Best Concentration (From Literature Sources)	Best Concentration (Concerted to Molar)	Variety	Age of Grapevine	Key Findings
1	Brassinosteroid	100 μmol L^−1^	100 μmol L^−1^	Kyoho	5-year	Exogenous BRs could change the aroma composition of grapes and increase the proportion of terpenes compounds such as α-pinene (0.03%), d-limonene (0.07%), and γ-terpene (63.67%) compared with the control group (0.01%, 0.04%, and 62.96%, respectively) [12].
2	Brassinosteroid	10 μmol L^−1^	10 μmol L^−1^	Shine Muscat	8-year	Exogenous BR plays an essential role in boosting the concentration of alcohols and aldehydes in eight-year-old “Shine Muscat” grape berries, as evidenced by a two-fold rise in 1-pentanol and sinapyl alcohol proportions than the control group after 48 h treatment. Additionally, the alcoholic compounds such as 1-hexanol and acetoin proportion (4.68% and 10.35%, respectively) were significantly more in content than the control group (2.92%, 5.73%, respectively). In the case of aldehydes, the proportions of methyl glyoxal (12.12%), pentanal (2.31%), and hexanal (22.96%) were found in abundance when compared with the control group (0%, 2.08%, and 15.27%, respectively) [13].

**Table 4 ijms-24-00445-t004:** Role of exogenous brassinosteroid/brassinolide to enhance the fruit quality parameter regarding molecular prospects.

S. No	Compounds	Best Concentration (From Literature Sources)	Best Concentration (Concerted to Molar)	Variety	Age of Grapevine	Key Findings
1	24-Epibrassinolide	0.4 mg L^−1^	0.83 μmol L^−1^	Cabernet Sauvignon	7-year	Exogenous EBL promoted the accumulation of total anthocyanins (~5 mg g^−1^) in grape berries compared to the control group (~2.5 mg L^−1^) and developed its pericarp color after 46 days treatment by up-regulating the expressions of genes *VvCHI1*, *VvCHS3*, *VvF3′5′H*, *VvDFR*, and *VvUFGT* (about 1.2–3 times increment) [43].
2	24-Epibrassinolide	0.4 mg L^−1^	0.83 μmol L^−1^	Cabernet Sauvignon	7-year	MYB5a, MYB5b, and MYBPA1, which interacted with anthocyanin biosynthesis structural genes. As seen in the late stage of grape berry ripening (120 days after anthesis), the EBL treatment (0.4 mg L^−1^) dramatically boosted the relative expressions of MYB5a (~0.5), MYB5b (~0.25), and MYBPA1 (~7) at 46 days after treatment as compared to control (~0.2, ~0.15, ~2, respectively) [43].
3	24-Epibrassinolide	0.4 mg L^−1^	0.83 μmol L^−1^	Cabernet Sauvignon	5-year	EBL treatment on grape increased the transcription levels of *VvHT3*, *VvHT4*, *VvHT5*, and *VvHT6* (monosaccharide-coding genes) at various stages of berry development, including half veraison stage and maturity stage, but had little effect on *VvHT1* and *VvHT2* expressions. The transcription level of *VvSUC27* (genes encoding disaccharide) was also considerably greater in grape berries after EBL treatment [44].
4	ABA, Brassinosteroid, ABA + Brassinoster-oid	10 μmol L^−1^	10 μmol L^−1^	Shine Muscat	8-year	The expression level of *VvHsfA2*, *VvGols1* and *VvHSP17.9* (stress resistance related genes) were all higher in BR-treated groups than that of in the control group (~1–1.5 times higher) [13].
5	24-Epibrassinolide	0.6 mg L^−1^	1.25 μmol L^−1^	Merlot	10-year	EBL significantly increased the expression of *VvSS* (Sucrose biosynthesis related gene) compared to the control group (around 2.75 times higher) [50].
6	24-Epibrassinosteroid	1.5 μmol L^−1^	1.5 μmol L^−1^	Shine Muscat	5-year	The application of EBR and 1.5 μmol L^−1^ on grape berries significantly promoted soluble solids accumulation by inhibition of *VvSKs*’ (glycogen biosynthesis related genes) expression [59].
7	24-Epibrassinolide	0.1 μmol L^−1^	0.1 μmol L^−1^	Cabernet Sauvignon	1-year	The expression of *GST* (pesticide degradation related gene) under EBL treatment was fourfold higher than that of the control group at third day after treatment and increased twofold at fifth day after treatment [56].
8	24-Epibrassinolide	0.1 μmol L^−1^	0.1 μmol L^−1^	Cabernet Sauvignon	1-year	The expression of *MRP* (pesticide degradation related gene) increased slightly in EBL treatment compared with the control group (~1.5 times higher). The transcription level of *P450* (pesticide degradation related gene) increased sharply to 4–6 times that of the control group after the application of EBL [56].

**Table 5 ijms-24-00445-t005:** Effect of exogenous brassinosteroid/brassinolide on fruit quality at enzymatic activity prospect.

S. No	Compounds	Best Concentration (From Literature Sources)	Best Concentration (Concerted to Molar)	Variety	Age of Grapevine	Key Findings
1	Brassinosteroid	100 μmol L^−1^	100 μmol L^−1^	Kyoho	5-year	The HMGR activity under exogenous BR treatment was significantly higher (10 times up-regulation) than the control group at the beginning and half veraison stages of grapevine [12].
2	Brassinosteroid	0.75 ppm	2.29 μmol L^−1^	Rish Baba	Not mentioned	Superoxide dismutase (SOD, ~50 U mg^−1^) activities of exogenous BR treatments were significantly higher than those of control group (~20 U mg^−1^, ~25 U mg^−1^, respectively) [37].
3	Brassinosteroid	1.5 ppm	4.58 μmol L^−1^	Rish Baba	Not mentioned	Exogenous BR grapes has significantly increased the levels of catalase (CAT, ~55 U mg^−1^) and peroxidase (POD, ~40 U mg^−1^) while 0.75 ppm treatment responsible to enhance the activity of ascorbate peroxidase (APX, ~35 U mg^−1^) than the control group [37].
4	24-Epibrassinolide	0.4 mg L^−1^	0.83 μmol L^−1^	Cabernet Sauvignon	5-year	EBL treatment in grape pericarp from days after application (DAA) 85 to 100, the glucose and fructose conversion enzymes “acidic invertase (INV) and neutral invertase (SuSyn)” significantly up-regulate their activity, while the 1.31 mg L^−1^ Brz (BR signaling inhibitor) application significantly reduced the acidic invertase (INV) activity and the INV activity at 60 DAA and 66 DAA, respectively [44].
5	24-Epibrassinolide	0.4 mg L^−1^	1.22 μmol L^−1^	Cabernet Sauvignon	5-year	Exogenous EBL significantly up-regulated the activities of key rate-limiting enzymes such as UDP-glucose: flavonoid 3-O-glucosyl transferase (UFGT: ~9 mmol g^−1^) and phenylalanine ammonia-lyase (PAL, ~240 U g^−1^) in comparison with the control group (~5 mmol g^−1^ and ~210 U g^−1^, respectively) [46].
6	24-Epibrassinolide	0.4 mg L^−1^	1.22 μmol L^−1^	Yan 73	5-year	Exogenous EBL treatment had higher UFGT (~23 mmol g^−1^) and PAL (~170 U g^−1^) activity than the control group (~18 mmol g^−1^ and ~150 U g^−1^, respectively) [46].
7	24-Epibrassinolide	0.4 mg L^−1^	0.83 μmol L^−1^	Redglobe	Not mentioned	The POD activity of grape berries treated with EBL (~30 U g^−1^, ~40 U g^−1^, ~36 U g^−1^, respectively) was significantly higher than that of control group at the 2nd, 4th, and 6th day (~8 U g^−1^, ~12 U g^−1^, ~28 U g^−1^, respectively) [40].
8	24-Epibrassinolide	0.8 mg L^−1^	1.66 μmol L^−1^	Redglobe	Not mentioned	The SOD activity of grape berries treated with EBL (~50 U g^−1^, ~58 U g^−1^, respectively) was significantly higher than that of control group at the 2nd and 6th day (~35 U g^−1^, ~44 U g^−1^, respectively) [40].
9	24-Epibrassinosteroid	0.1 μmol L^−1^	0.1 μmol L^−1^	Cabernet Sauvignon grape seedlings	Not mentioned	EBR on grape seedlings under drought stress significantly enhanced the level of APX activity after 12 h (~1.1 U g^−1^) and 24 h (~1.15 U g^−1^) of treatment as compared to control (stressed and unstressed). The EBR application was also responsible for enhancing the considerable activity of glutathione reductase (GR) at 48 h and 72 h treatment (~0.045 U g^−1^ and ~0.04 U g^−1^, respectively) compared with the drought stress treatment group (~0.03 U g^−1^ and ~0.035 U g^−1^, respectively) [47].
10	24-Epibrassinolide	0.6 mg L^−1^	1.25 μmol L^−1^	Merlot	Ten-year-old	Exogenous EBL enhanced the activity of sucrose phosphate synthase (SPS), significantly up-regulated activities of cell wall acid invertase (VvcwINV), sucrose transporter (VvSUC12), and sucrose synthase (VvSS) during veraison to ripening stage [50].
11	24-Epibrassinolide	3 μmol L^−1^ and 6 μmol L^−1^	3 μmol L^−1^ and 6 μmol L^−1^	Thompson Seedless	12-year	Both EBL applications (3 μmol L^−1^ and 6 μmol L^−1^) significantly promote the activities of polyphenol oxidase (PPO) with the value of ~12,000 U mg^−1^ and ~13,000 U mg^−1^, respectively). Interestingly, the content of total antioxidant activity (TAA) was significantly increased under 3 μmol L^−1^ EBL treatment (~4000 mmol 100 g^−1^) compared with the control group (~2000 mmol 100 g^−1^) [51].
12	24-Epibrassinolide	0.1 mg L^−1^	0.21 μmol L^−1^	Cabernet Sauvignon	1-year	After 12 h of EBL treatment, there was a substantial increase in superoxide dismutase (SOD, 13 U g^−1^ min^−1^) compared to the control group (9 U g^−1^ min^−1^) [52].
13	24-Epibrassinolide	0.1 μmol L^−1^	0.1 μmol L^−1^	Cabernet Sauvignon	1-year	EBL treatment (61.97 U mg^−1^ min^−1^) showed higher APX activity than the control group (37.33 U mg^−1^ min^−1^), and the average maximum increase of SOD activity was 25% higher than the control group [56].
14	Brassinosteroid	0.8 mg L^−1^	1.66 μmol L^−1^	Yan 73	5-year	The activity of PAL gradually rose, whereas the activity of UFGT increased and subsequently dropped. When compared to the control group, a concentration of exogenous BR significantly improved the activity of PAL (130 U g^−1^) and UFGT (28 U g^−1^) during the late stages of fruit ripening [79].
15	24-Epibrassinolide	0.8 mg L^−1^	2.44 μmol L^−1^	Summer Roya	Not mentioned	The EBL application was also responsible for enhancing the considerable activity of photosynthesis enzymes (35.16 SPAD) compared with the control group and the 0.4 mg L^−1^ EBL treatment group (28.12 SPAD, 33.53 SPAD, respectively) [83].

## Data Availability

All authors declared that data were contained in this manuscript.

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
