# Peer review of "Brassinosteroid Promotes Grape Berry Quality-Focus on Physicochemical Qualities and Their Coordination with Enzymatic and Molecular Processes: A Review"

_ijms, 2022, doi:10.3390/ijms24010445_

Round 1
Reviewer 1 Report
Comments and suggestions attached.

Author Response
[1] The classical research paper published in Plant Physiol (PMID: 16361521) on BR biosynthesis in grape and the influence of BR on ripening of grape berries should be included in the reference. One comprehensive review paper (PMID: 33643337) on plant hormone regulated heat stress tolerance during plant growth can be discussed and cited in section ‘4. The interaction of endogenous phytohormones…’ to increase the readability and visibility of the manuscript.
Thank you for your careful review and suggestions. According to your suggestions, we have added corresponding literature in section 4 (Lines 542-544, highlighted in yellow). I hope the above changes meet your expectations.
[2] The manuscript used both brassinolide and brassinosteroid, and abbreviated brassinolide as BR. The first discovered BR receptor was named BRI1 after BRASSINOSTEROID-IN-SENSITIVE 1. The authors should clarify the difference between brassinolide (BL) and brassinosteroid (BR) and use the terms consistently
Thanks for the highlights. According to your recommendation, we have clarified the difference between brassinolide (BL) and brassinosteroid (BR) (Line 28-30, highlighted in yellow) and use the terms consistently (references related to BR/EBR were highlighled in pueple, while references related to BL/EBL were highlighted in blue). We hope the modification could meet your requirement.
[3] Figure legends are too much simple. The legends must be understandable without reference to the text and include definitions of both symbols and abbreviations. In Figure 2, the symbol of dashed lines indicating the effect remains to be further discussed are misleading and can be changed or removed. Indicative terms for BRI1 are missing.
Thanks for your kind suggestions. As per your suggestion, we have enriched the Figure legend and highlighted it in yellow (Line 129-131; Line 132-136; Line 144-150; 563-566; Line 594-602, highlighted in yellow). At the same time, we deleted the wrong expression about the dashed lines you mentioned, and redraw the Figure 2. And added BRI1 protein to the diagram (Line 132-136, Figure 2).
[4] introduction, ‘External quality (fruit longitudinal…)’ vs ‘exterior (fruit expansion…)’; ‘internal quality (sugars…)’ vs ‘interior quality (sugars accumulation…)’ remove redundant terms and keep writing consistently.
Thanks for your careful review, based on your suggestions, we have harmonized ‘fruit interior quality (sugars accumulation…)’ to ‘fruit internal quality (sugars…)’, and harmonized ‘exterior quality (fruit expansion…)’ to ‘External quality (fruit longitudinal…)’ (Line 19-21; Line 38-41; Line 159-160; Line 608; highlighted in yellow).
[5] introduction, ‘aiming to provide a reliable reference for the producer to improve their quality and yield’ such a statement was far away from the aims of the writing and beyond the scope of the review.
Thanks for highlights. We have amended this sentence in the hope that the above changes will meet your requirements (Lines 24-25, highlighted in yellow).
[6]L32, ‘the sixth-largest’ what does it mean the largest?
As per your suggestion, we amended this sentence and hopefully that changes will meet your requirements (Line 33-36, highlighted in yellow).
[7]L37, ‘fruit longitudinal, transverse and diameter’
We are sorry for our errors in these sentences, the associated mis-presentation has been corrected and mentioned in the manuscript (Line 13; Line 19; Line 39; Line 159, highlighted in yellow).
[8]L53, ‘According to [12]’, correct this citation and others ([27]).
Thanks for correction regarding citation. We rechecked all the citation and correct them as you mentioned (Line 55; Line 197; Line 254; Line 352; Line 382; Line 447; Line 589, highlighted in yellow).
[9]L71, ‘Brassica nigra’
Thanks, we corrected this error and rewrote this sentence as well as highlighted in yellow (Line 74).
[10]L105, check the sentence.
Thanks for your reminder, we have checked this sentence and rephrase it (Lines 106-108, highlighted in yellow).
[11]L111, check the words and parentheses
Thanks for your reminder, we have checked the words and parentheses, and made corrections (Lines 112-114, highlighted in yellow).
Reviewer 2 Report
This review is devoted to the role of the phytohormone Brassinolide (BR) in fruit quality regulation, in particular grapes, associated with its biosynthesis and metabolism. This is a very good review and will certainly be of interest to readers of the special issue “Advanced Research of Plant Secondary Metabolism” of IJMC journal.
However, there are some points that, in my opinion, should be corrected before publication:
1) Line 32. The sentence “Therefore, it has been widely acknowledged to be the sixth-largest phytohormone, come after auxin (IAA), abscisic acid (ABA), cytokinin (CTK), gibberellin (GA), and 33 salicylic acid (SA).”
It seems to me wrong to write about auxins, gibberellins and cytokinins in the singular. These are groups of phytohormones that include several different structures. IAA (indole-3-acetic acid) is the designation for only one member of the auxin family. In addition, it is not entirely clear what is meant by the «largest»? By the size of the molecule? By molecular weight? This point needs to be clarified. I would also advise you to bring a Figure showing the structures of all the mentioned phytohormones.
2) In Table 1, all concentrations are given in different dimensions. I understand that these are data from various literature sources, but in this form, it is difficult to perceive and especially compare effective concentrations. It would be great, if possible, to add a column in which all values are converted to molar concentrations. Then, for example, it is easy to compare micromoles, nanomoles, picamoles, etc. But at the same time, it is necessary to keep the original literature values in another column with a footnote that these are the original literature data. The same advice for tables 2-5. This is a rather difficult and painstaking task, but if done, it will allow you to visually see the differences in concentrations at which a particular effect is achieved.
Author Response
1) Line 32. The sentence “Therefore, it has been widely acknowledged to be the sixth-largest phytohormone, come after auxin (IAA), abscisic acid (ABA), cytokinin (CTK), gibberellin (GA), and 33 salicylic acid (SA).”
It seems to me wrong to write about auxins, gibberellins and cytokinins in the singular. These are groups of phytohormones that include several different structures. IAA (indole-3-acetic acid) is the designation for only one member of the auxin family. In addition, it is not entirely clear what is meant by the «largest»? By the size of the molecule? By molecular weight? This point needs to be clarified. I would also advise you to bring a Figure showing the structures of all the mentioned phytohormones.
Thanks for your careful review and reminders. According to your suggestions, we have removed the wrong formulation of auxins (IAA), gibberellins (GA) and cytokinins (CTKs), and when this class of phytohormone appears, we state it in full terms (Line 34-36; Line 71; Line 72; Line 75-76; Line 84, highlighted in green). As well, we have restated IAA as indole-3-acetic acid (Line 66, highlighted in green), and re-clarify the meaning of ‘largest’ (Line 33-36, highlighted in green). Furthermore, we have complemented Figure 4, containing structures of all phytohormones involved in this manuscript (Line 562-566, highlighted in green).
2) In Table 1, all concentrations are given in different dimensions. I understand that these are data from various literature sources, but in this form, it is difficult to perceive and especially compare effective concentrations. It would be great, if possible, to add a column in which all values are converted to molar concentrations. Then, for example, it is easy to compare micromoles, nanomoles, picamoles, etc. But at the same time, it is necessary to keep the original literature values in another column with a footnote that these are the original literature data. The same advice for tables 2-5. This is a rather difficult and painstaking task, but if done, it will allow you to visually see the differences in concentrations at which a particular effect is achieved.
Thanks for your careful review and suggestions. According to your advice, we have unified the concentrations in Table 1-5, and hope that the above changes can make readers more intuitive to see the difference in the impact of BR dosage on fruit quality.